# Link Prediction or Perdition: the Seeds of Instability in Knowledge Graph Embeddings

## Abstract

Embedding models (KGEMs) constitute the main link prediction approach to complete knowledge graphs. Standard evaluation protocols emphasize rank-based metrics such as MRR or Hits@$K$, but usually overlook the influence of random seeds on result stability. Moreover, these metrics conceal potential instabilities in individual predictions and in the organization of embedding spaces. In this work, we conduct a systematic stability analysis of multiple KGEMs across several datasets. We find that high-performance models actually produce divergent predictions at the triple level and highly variable embedding spaces. By isolating stochastic factors, *i.e.*, initialization, triple ordering, negative sampling, dropout, hardware, we show that each independently induces instability of comparable magnitude. Furthermore, our results reveal no correlation between high MRR scores and stability. These findings highlight critical limitations of current benchmarking protocols, and raise concerns about the reliability of KGEMs for knowledge graph completion.

## 1 Introduction

Knowledge Graphs (KGs) (Hogan et al., 2021) serve as foundational structures for a wide range of applications in both academia and industry, including question answering (Yasunaga et al., 2021), semantic parsing (Berant et al., 2013), recommendation systems (Wang et al., 2018), among others (Hogan et al., 2021). They provide a formal and explicit network of facts in many downstream application domains (e.g., bio-chemistry, geography pharmacology, linguistics, etc.). In this perspective, KGs are used as a complement to LLMs, to supply reliable information to reason over, thereby improving grounding and potentially increasing explainability (Kau et al., 2024). KGs do so by encoding structured representations of information, typically defined within standard frameworks such as RDF (Lanthaler et al., 2014), where knowledge is encoded as triples $(h, r, t)$, indicating that a relation $r$ links a head entity $h$ to a tail entity $t$.

Despite their widespread adoption, real-world KGs are often incomplete due to their manual or semi-automatic construction process (Paulheim, 2017). For instance, in 2015, 58% of scientists in DBpedia lacked a link to their field of expertise (Krompaß et al., 2015). This incompleteness has motivated various refinement tasks (Hogan et al., 2021; Paulheim, 2017). Among them, *link prediction* (LP) aims to infer missing entities in incomplete triples of the form $(h, r, ?)$ or $(?, r, t)$.

Over the years, a variety of Knowledge Graph Embedding Models (KGEMs) have been proposed for LP and have now become the preponderant methods (Bordes et al., 2013; Yang et al., 2015; Dettmers et al., 2018; Ji et al., 2022). Being Machine Learning models, KGEMs should be evaluated following the good practice of reporting model performance by computing the average and standard deviation of results over multiple runs with identical configurations but different random seeds. This evaluation protocol allows to *(i)* exhibit average performance for fair benchmarking, and *(ii)* indicate model performance variance. In particular, a low variance suggests that retraining the model with different seeds yields similar results, implying that extensive hyperparameter tuning across seeds may be unnecessary.

However, in LP, this practice is seldom followed. Indeed, KGEMs are evaluated on reference benchmark datasets to identify the best performing ones (Ali et al., 2022), but results often only report the best run, and seeds may not be tuned. Even when seeds are considered, comparisons remain limited to global rank-based metrics such as Mean Reciprocal Rank (MRR) or Hits@$K$, which do not

Figure 1: Illustration of prediction variability across three runs of the same KGEM trained with different random seeds. While global rank-based metrics (e.g. MRR) may differ between runs (e.g., run 1 vs. runs 2 & 3) or match despite diverging predictions (run 2 vs. run 3), measuring similarity between top-$K$ predictions (e.g. Pred-Jaccard@5) reveals local stability differences.

capture local stability of predictions at the triple level. Consequently, two runs can exhibit similar aggregate scores while learning distinct embedding spaces and producing different triple-level predictions, ultimately yielding different completed graphs.

Figure 1 illustrates this phenomenon with the query (*Ribavirin, treatmentFor, ?*), where the goal is to rank the correct entity *Hepatitis* as high as possible among all candidates (excluding other known true entities). In run 1, the correct entity is ranked third, yielding a reciprocal rank of $0.33$, while in runs 2 and 3 it is ranked second, yielding $0.50$. In this toy example the MRR equals the reciprocal rank, but in practice it is averaged over many triples. This example is simple, yet it shows how the performance of a model can vary with the training seed.

Moreover, even when runs achieve the same performance, their predictions may diverge. For instance, runs 1 and 2 share four entities in the top-5 predictions (Jaccard similarity of $0.66$), whereas runs 2 and 3 share only one ($0.11$). This highlights the need to evaluate not only aggregate predictive capacity but also the stability of model predictions. Recent work confirms this concern, highlighting prediction-level divergence (*multiplicity*) (Marx et al., 2020; Zhu et al., 2024) as well as geometric instability of embeddings across seeds (Schumacher et al., 2021).

This observation raises a critical issue regarding the reliability of completed KGs via LP, as their content may vary depending on the random seed used during training. For instance, in Figure 1, completing the graph with the top-2 candidates for the query (*Ribavirin, treatmentFor, ?*) may yield *Asthma* and *Botulism*, *Asthma* and *Hepatitis*, or *Anemia* and *Hepatitis*, depending on the seed. This question is especially important for KGs supporting decision-making systems. This motivated our work, where we question and quantify the extent to which randomness affects both the predictive behavior and the internal representations of KGEMs. In particular, our research questions are as follows:

**RQ1.** Are KGEMs stable across different random seeds?

We conducted multiple runs of the same models while varying the sources of randomness. We measured both the similarity of predictions at the triple level, and the similarity of the learned embedding space. Our experimental analysis reveals that, despite apparent stability in aggregate metrics such as MRR, models exhibit pronounced variability at the triple level and in their embedding spaces.

**RQ2.** How do the different sources of randomness contribute to instability?

We isolated the contribution of each source of randomness, *i.e.*, initialization, triple ordering, negative sampling, and dropout, and showed that each individually induces instability. We identified hardware (*i.e.*, GPU) as a fifth source, producing variability of comparable magnitude.

**RQ3.** Is model stability correlated with link prediction performance?

We measured the stability of the best, median, and worst configurations identified in an hyperparameter search. While the worst-performing models are clearly highly unstable, we found no correlation between predictive performance and stability when comparing the best and median configurations: a higher MRR does not guarantee greater stability, neither in the embedding space nor in the predicted triples.

The remainder of the paper is organized as follows. In Section 2, we discuss prior work on KGEMs and their sensitivity to randomness. Section 3 describes our proposed formalism. The experimental protocol and results are reported in Section 4. We conclude with a summary of our findings and perspectives for future work in Section 5.

# 2 RELATED WORK

## 2.1 KNOWLEDGE GRAPH EMBEDDING MODELS FOR LINK PREDICTION

Knowledge Graph Embedding Models (KGEMs) encode KG entities and, in most cases, relations into a $d$-dimensional vector space with the aim of preserving as much as possible the structural regularities of the original KG while offering a compact encoding. In the context of link prediction, this transformation allows the model to capture implicit but statistically plausible patterns and facts, which can subsequently be used to infer new triples.

Link prediction with KGEMs is typically achieved by defining a scoring function $\varphi(h, r, t)$ that evaluates the plausibility of a triple $(h, r, t)$ based on the corresponding embedding vectors. The inferred facts can then complete the original KG. Over the past decade, a wide range of KGEMs has been proposed, which can be broadly grouped into three major families. **Geometric methods** define the scoring function as a distance measure between entities after applying a geometric transformation corresponding to the relation. This transformation can take various forms such as a translation in TransE (Bordes et al., 2013), a translation in a projected space for TransR (Lin et al., 2015), or a rotation for RotatE (Sun et al., 2019). **Tensor factorization methods** define the scoring function as a bilinear product capturing the similarity between entities under a linear transformation associated with the relation. The bilinear product can be restricted to a diagonal matrix as for DistMult (Yang et al., 2015), to a commuting normal matrix for ANALOGY (Liu et al., 2017), or extended to a complex embedding space for ComplEx (Trouillon et al., 2016). **Neural network approaches** learn the scoring function. Convolutional models such as ConvE (Dettmers et al., 2018) and ConvKB (Nguyen et al., 2018) use convolutional layers to capture patterns in embeddings, similarly to computer vision. Alternatively, Graph Neural Network models such as RGCN (Schlichtkrull et al., 2018) or more recently NBFNET (Zhu et al., 2021) exploit message passing to aggregate information from neighboring entities. Transformer-based approaches either operate solely on the KG structure (like all previous models) as for Hitter (Chen et al., 2021) or incorporate textual descriptions as for MoCoKGC (Li et al., 2024).

While these approaches differ in architecture and expressive power, they all share a common feature: their training process inherently involves randomness. In this work, we focus specifically on understanding the impact of such randomness on KGEM stability.

## 2.2 IMPACTS OF RANDOMNESS ON KNOWLEDGE GRAPH EMBEDDING MODELS

Machine Learning settings involve several factors that depend on randomness such as dataset splits, initialization of parameters, or dropout, with known impacts. In NLP, Madhyastha & Jain (2019) report that random seeds alter attention patterns and influential words, while, in GNNs, Shchur et al. (2019) show that data splits yield divergent performances. More generally, Marx et al. (2020) define the notion of multiplicity as the fact that similar global performance of models conceal substantial variation at the local level due to randomness factors. They also propose two metrics to quantify this phenomenon: *Discrepancy* and *Ambiguity*.

In the KGEM literature, the impact of randomness during training is seldom addressed. Section 4.3 of (Bonner et al., 2022) studies the effect of initialization seeds and concludes that most models are robust as global metrics such as MRR exhibit low variance. In contrast, Zhu et al. (2024) adapt the notion of multiplicity to KGEMs and show that models retrained with different seeds may produce markedly different individual predictions at the triple level. They demonstrate that this instability can be alleviated with ensemble voting. However, they only assess model agreement on the presence or absence of the gold truth entity in the top 10 candidates and disregard other predicted entities.

Beside the stability of predictions, other works investigated the impacts of randomness on the embedding space itself. For example, Schumacher et al. (2021) compare the topologies of learned embedding spaces across seeds and report substantial instability, leading to instable individual pre-

dictions even if global accuracy remain stable. Nonetheless, their work only focuses on embedding models for homogeneous graphs. In the KGEM literature, Jain et al. (2021) and Hubert et al. (2024) investigate properties of embedding spaces, such as the (non-)preservation of class membership and the preservation of graph proximity. But, to the best of our knowledge, no prior work has examined the stability of these spaces with respect to randomness.

This state of the art, and the need to evaluate the stability of link prediction for real applications, motivated us to quantify the impact of randomness on the organization of knowledge graph embedding spaces, and its effect on predictions of KGEMs. Additionally, we investigate prediction stability not only considering the presence or absence of the gold truth entity, but taking into account all top-$K$ predicted candidates. KGEM stability is a major issue since it may lead to unreliable and seed-dependent KG completion, affecting the trustworthiness of completed KGs as a grounding structure and their usefulness in applications supporting decision-making.

## 3 Measuring KGEM (In)stability

### 3.1 Preliminaries and Notations on KGEM Training

In our work, we consider the formal representation of a Knowledge Graph (KG) as a set of entities $\mathcal{E}$, a set of relations $\mathcal{R}$, and a collection of triples $\mathcal{T} \subseteq \mathcal{E} \times \mathcal{R} \times \mathcal{E}$ encoding facts. Each triple $(h, r, t)$ consists of a head entity $h$, a relation $r$, and a tail entity $t$, e.g., (France, hasCapital, Paris).

We focus on Knowledge Graph Embedding Models (KGEMs) trained for Link Prediction (LP). In this case, $\mathcal{T}$ is split into train ($\mathcal{T}_{\text{train}}$), validation ($\mathcal{T}_{\text{val}}$), and test ($\mathcal{T}_{\text{test}}$) sets, the latter being used for model evaluation. Given a test triple $(h, r, t) \in \mathcal{T}_{\text{test}}$, LP consists in predicting missing entities in $(h, r, ?)$ or $(?, r, t)$ by scoring and ranking all entities of the KG and expecting the gold-truth entity to be as highly ranked as possible. Model quality is therefore assessed over $\mathcal{T}_{\text{test}}$ with rank-based metrics such as Mean Rank (MR), Mean Reciprocal Rank (MRR), and Hits@$K$ (typically $K \in \{1, 3, 10\}$).

Several families of KGEMs exist (Section 2.1), and their training typically involves hyperparameter tuning to obtain a **hyperparameter configuration $C$** that maximizes validation MRR or Hits@$K$.

A less explored aspect lies in the impacts of randomness during training, associated with the choice of random seeds. In this work, we investigate the impact of such random factors on model results and performance. We identify four independent sources of randomness:

**Negative sampling ($\mathcal{N}$)** random corruption of positive triples to generate negative examples.

**Triple ordering ($\mathcal{O}$)** random shuffling of training triples at each epoch.

**Parameter initialization ($\mathcal{I}$)** random initialization of model parameters.

**Dropout randomness ($\mathcal{D}$)** random subset of neurons deactivated at each batch.

We define a **seed configuration** $\mathfrak{S}$ as a tuple that assigns a specific value to each source of randomness. For example, $\mathfrak{S} = \{\mathfrak{S}_{\mathcal{N}} = s_a, \ \mathfrak{S}_{\mathcal{O}} = s_b, \ \mathfrak{S}_{\mathcal{I}} = s_c, \ \mathfrak{S}_{\mathcal{D}} = s_d\}$ sets the seed $\mathfrak{S}_{\mathcal{N}}$ controlling negative sampling to $s_a$, and the other components follow analogously.

An experimental **instance** $I$ is defined as $I = \text{Instance}(\text{Dataset}, \text{KGEM}, C, \mathfrak{S}, \mathcal{H})$, indicating that a KGEM is trained on the specified Dataset with hyperparameter configuration $C$, seed configuration $\mathfrak{S}$, and on hardware $\mathcal{H}$. In our experiments, we fix $\mathcal{H}$ to ensure identical execution conditions across all runs, and therefore omit it from the notation. We also omit the Dataset and KGEM when clear from context, and thus denote instances compactly by $I_{\mathfrak{S}}^{C}$.

Let $S = \{s_1, \ldots, s_n\}$ be a set of seed values. We define a **comparison group** $G$ as a set of instances obtained by varying one seed of $\mathfrak{S}$ over $S$, while keeping the others fixed to $s_1$. For example, the group of instances where only negative sampling varies is:

$$G_{\mathcal{N}}^{C} = \left\{ I_{\{\mathfrak{S}_{\mathcal{N}} = s_i, \ \mathfrak{S}_{\mathcal{O}} = s_1, \ \mathfrak{S}_{\mathcal{I}} = s_1, \ \mathfrak{S}_{\mathcal{D}} = s_1\}}^{C} \ \middle| \ s_i \in S \right\}$$

We define $G_{\mathcal{I}}^{C}$, $G_{\mathcal{O}}^{C}$, $G_{\mathcal{D}}^{C}$ similarly for the other sources of randomness, and $G_{\mathcal{A}}^{C}$ for the case where all seeds vary jointly over $S$. Our objective is to determine whether predictions and embedding

space organization of instances within the same group remain consistent, using spatial and prediction metrics introduced in the next section.

## 3.2 METRICS

### 3.2.1 JACCARD SIMILARITY ON PREDICTIONS AND EMBEDDING SPACE NEIGHBORS

To assess the instability of a group $G_{\mathcal{X}}^C$ for any source $\mathcal{X}$ of randomness, we rely on similarity metrics that capture consistency either at the prediction level or in the embedding space. We use Jaccard similarity as our main baseline due to its simplicity, adequacy and interpretability. Given two sets $A$ and $B$, the Jaccard similarity is defined as follows:

$$\text{Jaccard}(A, B) = \frac{|A \cap B|}{|A \cup B|} \tag{1}$$

This measure quantifies the overlap between two unordered sets, ranging from 0 (disjoint sets) to 1 (identical sets). We apply Jaccard similarity both on the $K$ top predictions (Pred-Jaccard@$K$), and $K$ nearest neighbors in the embedding space (Space-Jaccard@$K$), as detailed below.

**Pred-Jaccard@$K$.** KG completion with Link Prediction can be assumed to fill missing facts by directly adding the top-$K$ predicted triples into the KG for each incomplete triple $(h, r, ?)$ or $(?, r, t)$, in line with usual evaluation metrics such as Hits@$K$. Pred-Jaccard@$K$ thus measures the extent to which two instances of a model would insert the same triples to the KG. Let us denote $\mathcal{Q}_{\text{test}}$ all the test queries that can be derived from test triples, *i.e.*, queries $(h, r, ?)$ and $(?, r, t)$ for each test triple $(h, r, t)$. Let $\nu(I, q, K)$ be the top-$K$ predictions of instance $I$ for query $q$. Considering two model instances $I_1$ and $I_2$, we define Pred-Jaccard@$K$ as follows:

$$\text{Pred-Jaccard@}K(I_1, I_2) = \frac{1}{|\mathcal{Q}_{\text{test}}|} \sum_{q \in \mathcal{Q}_{\text{test}}} \text{Jaccard}\big(\nu(I_1, q, K), \nu(I_2, q, K)\big) \tag{2}$$

A high score (*i.e.*, close to 1) indicates that model instances $I_1$ and $I_2$ would enrich the KG with largely overlapping triples.

**Space-Jaccard@$K$.** We also seek to measure the similarity between embedding spaces $E_1$ and $E_2$ of the entities $\mathcal{E}$ learned by two instances $I_1$ and $I_2$. Embedding spaces are not directly comparable, since they may differ by various transformations, *e.g.*, rotation. Rather than aligning embeddings, we assess stability through neighborhood preservation. For each entity $e_i \in \mathcal{E}$, let $\mathcal{N}_K^{E_j}(e_i)$ denote its $K$ nearest neighbors in embedding space $E_j$. We define Space-Jaccard@$K$ as follows:

$$\text{Space-Jaccard@}K(E_1, E_2) = \frac{1}{|\mathcal{E}|} \sum_{e_i \in \mathcal{E}} \text{Jaccard}\big(\mathcal{N}_K^{E_1}(e_i), \mathcal{N}_K^{E_2}(e_i)\big) \tag{3}$$

This metric reflects the extent to which local neighborhoods are consistently preserved across embedding spaces of different instances.

**Aggregation over groups.** To obtain a stability score for a group $G_{\mathcal{X}}^C$, we compute the above metrics across all pairs $(I_1, I_2) \in G$, and report the mean and standard deviation.

### 3.2.2 OTHER METRICS.

We also tested alternative similarity measures, beyond Jaccard. For predictions, we considered Rank-Biased Overlap (RBO), which operates on ranked lists and emphasizes agreement among top-ranked elements. We also measured Ambiguity and Discrepancy (Zhu et al., 2024), which quantify the level of agreement between models with respect to the gold truth. For embedding spaces, we considered Centered Kernel Alignment (CKA) (Kornblith et al., 2019) widely used in machine learning. A detailed description of these metrics and a discussion of their application to our experimental setting is provided in Appendix C. Crucially, all these measures confirm the same instability picture, described below.

Table 1: Dataset statistics.

| Dataset | # Entities | # Relations | # Edges | Mean in-degree |
|---|---|---|---|---|
| WN18RR | 40,943 | 11 | 93,003 | 2.72 |
| Kinship | 104 | 25 | 10,686 | 82.15 |
| Nations | 14 | 55 | 1,992 | 113.71 |
| CoDEx-S | 2,034 | 42 | 36,543 | 32.53 |

## 4 EXPERIMENTATION

### 4.1 EXPERIMENTAL SETTING

**Models.** We select representative KGEM from different families: **TransE** (Bordes et al., 2013) (geometric-based), **DistMult** (Yang et al., 2015) (tensor factorization method), **ConvE** (Dettmers et al., 2018) (convolutional neural networks), **RGCN** (Schlichtkrull et al., 2018) (graph neural networks), and a **Transformer** without neighborhood context from Chen et al. (2021) (transformer networks).

**Datasets.** We evaluate our models on the following standard KG datasets from the literature: **WN18RR** (Dettmers et al., 2018) a lexical dataset subset of WordNet, known for its hierarchical structure; **CoDEx-S** (Safavi & Koutra, 2020) a subset of Wikidata, containing real-world facts; and **Kinship** and **Nations**, smaller datasets adopted in the literature and respectively describing familial and political relationships. Datasets statistics are summarized in Table 1.

In our experiments, the set $S$ of seed values has been fixed to $\{42, 283, 358, 698, 887\}$, and the hardware $\mathcal{H}$ has been fixed to a GPU *Nvidia GeForce RTX 2080 Ti (11 GiB)*.

Our code[1] and learned embeddings[2] are publicly available. Our code allows full control and independent manipulation of the four sources of randomness.

**Hyperparameter Search.** The purpose of this tuning step is not to identify the optimal model configuration, a task that would require exploring a substantially larger search space (Ruffinelli et al., 2020), but rather to provide a meaningful spectrum of models with different LP performance levels to answer RQ3. As shown in Appendix A, the best configurations we identified achieve performance within a range comparable to state-of-the-art results.

Accordingly, we fixed the seed configuration to $\mathfrak{S} = \{\mathfrak{S}_{\mathcal{N}} = s_1, \ \mathfrak{S}_{\mathcal{O}} = s_1, \ \mathfrak{S}_{\mathcal{I}} = s_1, \ \mathfrak{S}_{\mathcal{D}} = s_1\}$ and conducted a compact hyperparameter search over embedding dimensions $\{128, 256, 512\}$ and learning rates $\{10^{-6}, 10^{-5}, 10^{-4}, 10^{-3}, 10^{-2}, 10^{-1}\}$. All models are initialized using Xavier normal, and we employ the cross-entropy loss, as Ruffinelli et al. (2020) shows that this choice consistently yields reliable results. Dropout rates for entities and relations are fixed at 0.2, the batch size is set to 256, and the number of negative samples is dataset-dependent: 10 for Kinship and Nations, and 500 for all other datasets. More details are provided in Appendix A.

We obtained three representative model configurations: the best (highest MRR) performing model $C_B$, the worst (lowest MRR) performed model $C_W$, and the median performing model $C_M$, chosen among those achieving a MRR greater than 0.05.

### 4.2 RESULT ANALYSIS

#### 4.2.1 RQ1: STABILITY W.R.T. RANDOMNESS

If we only consider aggregate scores such as MRR, KGE models appear to be remarkably stable. Table 2 reports the mean MRR and standard deviation across five runs in $G_{\mathcal{A}}^{C_B}$ (i.e., the best model with all seed components varying jointly). Deviations appear consistently small, suggesting that models trained under the same hyperparameter configuration but with varying seeds still achieve

---

[1]Provided in the Supplementary Material and to be released on GitHub upon acceptance.

[2]Will be made available on Zenodo upon acceptance.

Table 2: Mean and standard deviation of MRR over five runs of the best model configuration, with all seeds jointly varying over $S$.

| Dataset | ConvE | TransE | DistMult | RGCN | Transformer |
|---|---|---|---|---|---|
| WN18RR | $0.410 \pm 0.003$ | $0.194 \pm 0.002$ | $0.422 \pm 0.001$ | $0.389 \pm 0.001$ | $0.262 \pm 0.005$ |
| Kinship | $0.802 \pm 0.006$ | $0.214 \pm 0.001$ | $0.658 \pm 0.009$ | $0.585 \pm 0.003$ | $0.806 \pm 0.009$ |
| Nations | $0.792 \pm 0.007$ | $0.502 \pm 0.003$ | $0.785 \pm 0.007$ | $0.658 \pm 0.066$ | $0.661 \pm 0.017$ |
| CoDEx-S | $0.434 \pm 0.003$ | $0.348 \pm 0.001$ | $0.413 \pm 0.003$ | $0.362 \pm 0.010$ | $0.360 \pm 0.006$ |

comparable overall predictive performance measured by the aggregated ranks of the gold-truth entities.

However, stability in aggregate scores does not imply convergence to similar local states. Comparing learned embedding spaces (Figure 2a) reveals substantial discrepancies across runs; for instance, on WN18RR, Space-Jaccard@10 remains below 0.5 for all models except TransE. This indicates that distinct latent organizations can still produce comparable global performance.

Variability also arises at the prediction level: triple-by-triple comparisons (Figure 2b and 2c) reveal that different seeds lead to different predictions for individual test triples. For example, on WN18RR with RGCN, the Pred-Jaccard@1 is 0.59. Consider a completion policy that adds to the KG each test triple completed with the top-ranked entity predicted by the model. Such a Pred-Jaccard@1 value indicates that, on average, about 41% of the triples added to the KG will differ across runs, leading to divergent KGs which may undermine the trustworthiness of downstream decision-making processes.

These findings point to a persistent form of instability across all datasets. Some models are nevertheless more robust than others. In particular, TransE exhibits the highest stability, with Pred-Jaccard@1 consistently above 0.8 across datasets. At the opposite end, RGCN and Transformer show pronounced instability, especially in the embedding space. We conjecture that the larger number of parameters of these architectures grants them more degrees of freedom, thereby amplifying divergence across runs.

Variability both in space (Space-Jaccard@10) and in predictions (Pred-Jaccard@10) indicates that models learn different latent spaces, and thus encode different knowledge. However, this variability is not reflected in LP performance when considering only the ranks of gold entities (MRR), despite potentially leading to divergent completed KGs. This observation calls for additional performance perspectives and metrics, beside MRR, to provide a more comprehensive assessment of KGEM performance for KG completion, especially for KG-based decision-making applications.

### 4.2.2   RQ2: IMPACT OF INDIVIDUAL SOURCES OF RANDOMNESS

We now examine the impact of each source of randomness in isolation. In particular, we compare stability across $G_{\mathcal{N}}^{C_B}$, $G_{\mathcal{I}}^{C_B}$, $G_{\mathcal{O}}^{C_B}$, $G_{\mathcal{D}}^{C_B}$, and $G_{\mathcal{A}}^{C_B}$. Figure 3a reports the impact on embedding space stability on WN18RR, while Figure 3b shows the corresponding effect on triple predictions. Results on other datasets are available in Appendix B. Our results indicate that each source independently is sufficient to induce instability at both the embedding and prediction levels. This highlights the inherent and persistent instability induced by randomness factors in model training. Across all datasets and metrics (see Figure 5), no particular seed emerges as a dominant factor of instability. The observed variations in instability appear to be more strongly tied to the model architecture than to the specific source of randomness.

Even more strikingly, we define an additional group $G_{\mathcal{H}}^{C_B,\mathfrak{S}}$, in which the hardware varies while keeping both the hyperparameter configuration $C_B$ and seed configuration $\mathfrak{S}$ fixed. We run models on multiple GPUs: *Nvidia GeForce RTX 2080 Ti (11 GiB)*, *Nvidia GeForce GTX 1080 Ti (11 GiB)*, *Nvidia Tesla T4 (15 GiB)*, *Nvidia A40 (45 GiB)*, and *Nvidia A100-SXM4-40GB (40 GiB)*. Unexpectedly, this setting exhibits instability of the same magnitude as seed variation, highlighting the challenge of ensuring reproducibility both in research and production environments, particularly in cloud computing scenarios where hardware choice may be beyond the user's control.

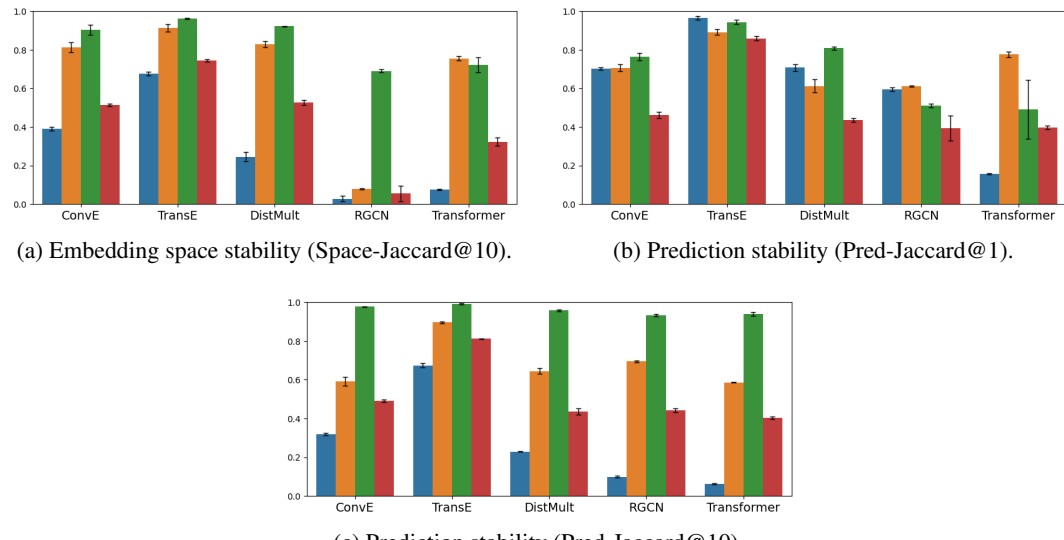

(a) Embedding space stability (Space-Jaccard@10).

(b) Prediction stability (Pred-Jaccard@1).

(c) Prediction stability (Pred-Jaccard@10).

Figure 2: Stability analysis across all datasets and models. Scores on Nations (and Kinship) are inflated, especially for Jaccard@10 due to the small number of entities (14 and 104, respectively). The color codes represent the datasets: ■ WN18RR ■ Kinship ■ Nations ■ CoDEx-S

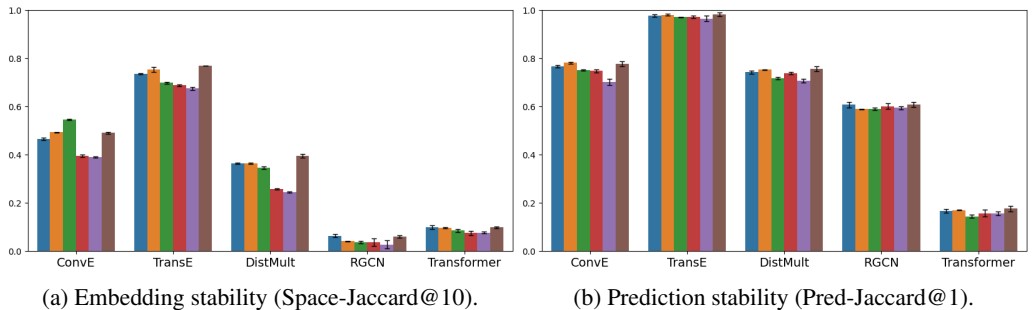

(a) Embedding stability (Space-Jaccard@10).

(b) Prediction stability (Pred-Jaccard@1).

Figure 3: Impact of individual sources of randomness on WN18RR. Each source of randomness (including hardware) is varied independently while keeping all others fixed. Colors indicate the corresponding source of randomness: ■ seed_init ■ seed_dropout ■ seed_neg ■ seed_order ■ all ■ hardware. The category "all" includes all sources of randomness but hardware.

### 4.2.3 RQ3: Relationship Between Stability and Link Prediction Performance

We now investigate the link between predictive performance, as measured by the MRR metric, and stability of KGEMs. In other words, we aim to identify which of $G_A^{C_B}$, $G_A^{C_M}$, and $G_A^{C_W}$, *i.e.*, respectively the best, medium, and worst configurations from the hyperparameter search, exhibits the greatest instability.

Our observations (Figures 4a and 4b) show that poorly performing models tend to be less stable, which aligns with intuition. When excluding the small-scale datasets Kinship and Nations which are too limited to provide generalizable insights, we observe no clear relationship between model performance and stability between high and median-performing models. At both the embedding and prediction levels, high-performing models do not consistently exhibit greater stability than median-performing ones. This suggests that, for sufficiently large datasets, stability is independent of model predictive performance and, instead, may be more strongly influenced by factors such as model architecture and hyperparameter choices (*e.g.*, learning rate and embedding dimension)[3].

---

[3]It is also worth noting that in our hyperparameter tuning (Section 4.1 and Appendix A), we did not vary dropout, which may increase MRR while simultaneously decrease stability.

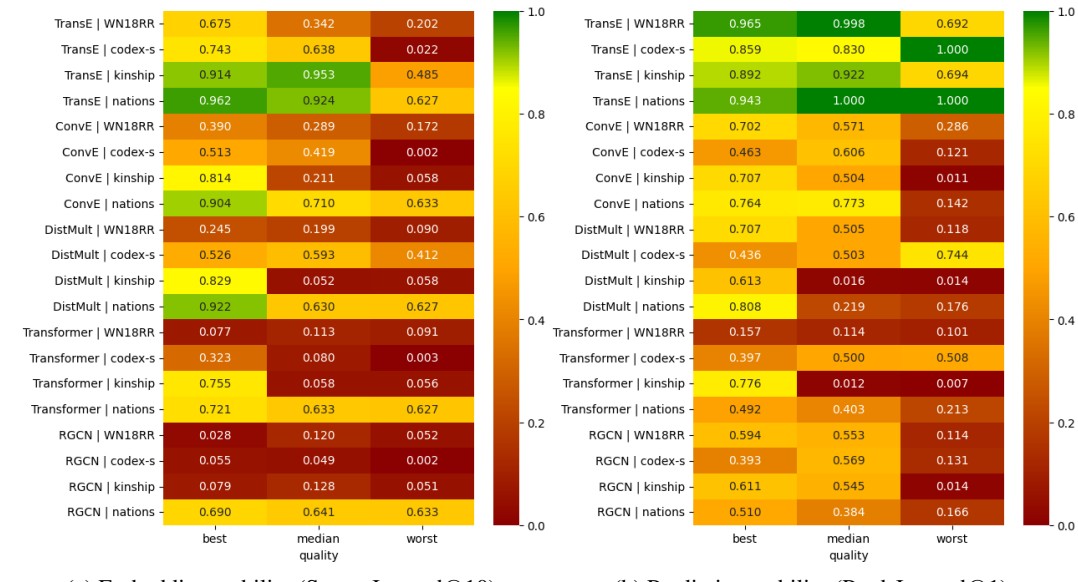

(a) Embedding stability (Space-Jaccard@10).  (b) Prediction stability (Pred-Jaccard@1).

Figure 4: Effect of model performance on stability.

## 5 CONCLUSION

We conducted a systematic stability analysis of several KGEMs across multiple datasets. After hyperparameter optimization, models were retrained with different random seeds. Although aggregate metrics such as MRR and Hits@$K$ appeared stable, they hid pronounced instabilities. At the embedding level, these instabilities emerged as low Jaccard similarity between entity neighborhoods across seeds. At the prediction level, model predict divergent top-$K$ candidates for identical triples.

To investigate the sources of instability, we disentangled stochastic factors such as initialization, triple ordering, negative sampling, and dropout, and showed that each induces instability of comparable magnitude. This result validates the implementation choice of frameworks such as PyKeen and LibKGE to only allow a unique seed value for all factors. Even hardware differences (i.e., GPU type) contributed to variability, underscoring the difficulty of ensuring reproducibility in both research and production settings. Our observations thus also call for openly publishing embeddings alongside source code to ensure research reproducibility.

Our results also complement prior work (Jain et al., 2021; Hubert et al., 2024), which cautioned that embeddings do not necessarily preserve semantics or graph proximity. We show that latent neighborhoods are themselves partly seed-dependent, underscoring the risks of treating KGEMs as faithful neural counterparts of KGs. This instability also questions the validity of downstream operations on the embedding space such as similarity measurement, clustering, neighborhood-based or space-based reasoning (Monnin et al., 2022; Xiong et al., 2023; Zhang et al., 2019).

At the prediction level, instability raises serious concerns for KG completion: results may vary across seeds, leading to unreliable KG augmentation. Such variability undermines the trustworthiness of completed KGs as grounding structures and limits their reliability in downstream applications that support decision-making. We also demonstrated that performance and stability are not aligned criteria for model selection, highlighting the need to treat them as distinct evaluation dimensions. These findings advocates for additional metrics and perspectives to comprehensively measure the performance of KGEMs on link prediction settings, not only considering academic benchmarks but also application requirements.

While voting strategies have been proposed to mitigate multiplicity (Zhu et al., 2024), they incur substantial computational cost. Our results further indicate that certain models, such as TransE, exhibit comparatively higher stability. This observation opens a promising direction for future work: the design of novel architectures or training protocols (e.g., stability-aware loss functions) that explicitly enforce stability.

## ETHICS STATEMENT

This work makes use of publicly available datasets, such as subsets of Wikidata and Wordnet. While these datasets contain information originally created by humans, they only include publicly accessible and common knowledge, without sensitive or private data. As such, our study does not raise concerns regarding privacy, consent, or discrimination. No institutional review board (IRB) approval was required. We believe that our methodology and results do not pose risks of harmful applications.

## REPRODUCIBILITY STATEMENT

We have taken extensive measures to ensure the reproducibility of our findings. All code is provided anonymously in the supplemental materials, together with scripts to reproduce training, evaluation, and similarity computations. Hyperparameter configurations, grid search outcomes, and random seeds are documented in the appendix. We report the hardware used (GPUs) and explicitly state the seeds employed in each experiment. The datasets involved are public and, for smaller ones such as *Kinship* and *Nations*, can be easily reused to replicate our results. Furthermore, should the paper be accepted, we will release all pretrained embeddings and prediction results on `Zenodo`, ensuring long-term availability and independent verification.

## LLM USAGE

In the preparation of this paper, large language models were used as a writing assistant. Specifically, LLMs were employed to refine the English phrasing of certain paragraphs. They also supported bibliographic exploration outside the field of knowledge graph embeddings through deep search. In addition, LLMs were used as coding support for implementation details. All scientific contributions, analyses, and experimental results presented in this paper are the authors' own.

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

# APPENDICES

## A HYPERPARAMETERS

We fixed the seed configuration to $\mathfrak{S} = \{\mathfrak{S}_{\mathcal{N}} = s_1, \ \mathfrak{S}_{\mathcal{O}} = s_1, \ \mathfrak{S}_{\mathcal{I}} = s_1, \ \mathfrak{S}_{\mathcal{D}} = s_1\}$ and conducted a compact hyperparameter search over embedding dimensions $\{128, 256, 512\}$ and learning rates $\{10^{-6}, 10^{-5}, 10^{-4}, 10^{-3}, 10^{-2}, 10^{-1}\}$. Table 3 summarizes the corresponding configurations.

All models are initialized using Xavier normal, and we employ the cross-entropy loss, as Ruffinelli et al. (2020) shows that this choice consistently yields reliable results. Dropout rates for entities and relations are fixed at $0.2$, the batch size is set to $256$, and the number of negative samples is dataset-dependent: 10 for Kinship and Nations, and 500 for all other datasets. Inverse relations are enabled by default: for each training triple $(h, r, t)$, we additionally generate $(t, r^{-1}, h)$, and during inference, queries of the form $(?, r, t)$ are replaced by $(t, r^{-1}, ?)$.

For TransE, we adopt the $\ell_2$ norm, in accordance with the original paper (Bordes et al., 2013).

For ConvE, we set the projection dropout to $0.3$ and the feature map dropout to $0.2$, following both the LibKGE configuration[4] and the original implementation[5].

For Transformer, we adopt the LibKGE configuration[6], which specifies 8 attention heads, a feedforward dimension of $1280$, 3 encoder layers, ReLU activation, and an encoder dropout of $0.1$.

For RGCN, we apply a hidden dropout of $0.2$ and use two encoding layers, following the PyTorch Geometric RGCN example[7]. We employ basis decomposition with two basis functions, as done in the original paper for FB15k and WN18 (Schlichtkrull et al., 2018).

Table 3: Learning rate and embedding dimension for each model, dataset, and configuration.

| Model | Dataset | Best | Median | Worst |
|---|---|---|---|---|
| Transformer | CoDEx-S | (0.1, 256) | (0.01, 512) | (0.0001, 128) |
| | WN18RR | (0.01, 512) | (0.1, 128) | (0.01, 128) |
| | Kinship | (0.1, 128) | (0.001, 512) | (0.1, 512) |
| | Nations | (0.1, 256) | (0.0001, 128) | (1e-06, 256) |
| TransE | CoDEx-S | (0.01, 512) | (0.001, 512) | (1e-05, 512) |
| | WN18RR | (0.01, 512) | (0.1, 256) | (0.001, 128) |
| | Kinship | (0.1, 128) | (0.1, 512) | (0.001, 128) |
| | Nations | (0.1, 512) | (0.0001, 512) | (1e-06, 256) |
| DistMult | CoDEx-S | (0.1, 128) | (0.01, 512) | (0.001, 128) |
| | WN18RR | (0.01, 512) | (0.001, 512) | (0.001, 128) |
| | Kinship | (0.01, 128) | (0.0001, 256) | (1e-06, 512) |
| | Nations | (0.1, 128) | (0.0001, 512) | (1e-06, 256) |
| ConvE | CoDEx-S | (0.001, 512) | (0.001, 128) | (1e-06, 256) |
| | WN18RR | (0.001, 512) | (0.001, 256) | (0.001, 128) |
| | Kinship | (0.1, 256) | (0.0001, 512) | (1e-06, 128) |
| | Nations | (0.01, 512) | (0.001, 128) | (1e-06, 128) |
| RGCN | CoDEx-S | (0.1, 128) | (0.001, 128) | (1e-05, 256) |
| | WN18RR | (0.1, 256) | (0.01, 256) | (0.001, 128) |
| | Kinship | (0.01, 128) | (0.01, 512) | (1e-06, 256) |
| | Nations | (0.1, 128) | (0.0001, 256) | (1e-06, 128) |

---

[4]https://github.com/uma-pi1/kge/blob/master/kge/model/conve.yaml
[5]https://github.com/TimDettmers/ConvE/blob/master/main.py
[6]https://github.com/uma-pi1/kge/blob/master/kge/model/transformer.yaml
[7]https://github.com/pyg-team/pytorch_geometric/blob/master/examples/rgcn_link_pred.py#L42

Table 4: Comparison of MRR for TransE, ConvE, DistMult, RGCN, and Transformer (HittER, no-context) between SOTA and our results. SOTA results on WN18RR are taken from LIBKGE (Ruffinelli et al., 2020); CoDEx-S from the original CoDEx paper (Safavi & Koutra, 2020); Transformer from its original paper (Chen et al., 2021); Kinship and Nations from the Py-KEEN repositories (see notes). When not reported, values are indicated with "–".

| | ConvE | TransE | DistMult | RGCN | Transformer |
|---|---|---|---|---|---|
| **SOTA** | | | | | |
| WN18RR | 0.442 | 0.228 | 0.452 | – | 0.473 |
| CoDEx-S | 0.444 | 0.354 | – | – | – |
| Kinship | 0.835 | 0.444 | 0.636 | – | – |
| Nations | – | 0.403 | – | – | – |
| **Ours** | | | | | |
| WN18RR | $0.410 \pm 0.003$ | $0.194 \pm 0.002$ | $0.422 \pm 0.001$ | $0.389 \pm 0.001$ | $0.262 \pm 0.005$ |
| CoDEx-S | $0.434 \pm 0.003$ | $0.348 \pm 0.001$ | $0.413 \pm 0.003$ | $0.362 \pm 0.010$ | $0.360 \pm 0.006$ |
| Kinship | $0.802 \pm 0.006$ | $0.214 \pm 0.001$ | $0.658 \pm 0.009$ | $0.585 \pm 0.003$ | $0.806 \pm 0.009$ |
| Nations | $0.792 \pm 0.007$ | $0.502 \pm 0.003$ | $0.785 \pm 0.007$ | $0.658 \pm 0.066$ | $0.661 \pm 0.017$ |

Recall that our goal is not to identify hyperparameters that achieve state-of-the-art performance, but rather to derive three distinct performance levels. As shown in Table 4, the best configuration we identified yields performance that falls within a comparable range to SOTA results.

## B ALL DATASETS OF RQ2

Figure 5 summarizes our results on all datasets and confirms our analysis in Section 4.2.2:

- each source is independently sufficient to induce instability at both the embedding and prediction levels;

- there is an inherent and persistent instability induced by randomness factors in model training;

- the observed variations in instability appear to be more strongly tied to the model architecture than to the specific source of randomness.

## C OTHER METRICS

### C.1 RANK-BIASED OVERLAP (RBO)

While Jaccard similarity only considers set overlap, it ignores ranking information within the top-$K$ elements. Rank-Biased Overlap (RBO) addresses this limitation by assigning higher weight to higher-ranked elements, thereby emphasizing agreement on the top of the list. We adopt the standard setting with persistence parameter $p = 1$, which corresponds to truncation at depth $K$.

Formally, for two ranked lists $A$ and $B$, RBO at depth $K$ is defined as:

$$\text{RBO@}K(A, B) = \frac{1}{K} \sum_{d=1}^{K} \frac{|A_{1:d} \cap B_{1:d}|}{d}, \tag{4}$$

where the numerator counts the number of common items among the top-$d$ elements of $A$ and $B$. A score of 1 means identical top-$K$ orderings, whereas a score of 0 means no overlap.

---

[7]https://github.com/pykeen/benchmarking

[7]https://github.com/pykeen/ranking-metrics-manuscript/tree/main/models/nations

[7]RGCN was originally benchmarked on earlier datasets FB15k and WN18 with train-test leakage issues. No subsequent re-evaluations on the considered benchmarks have been found.

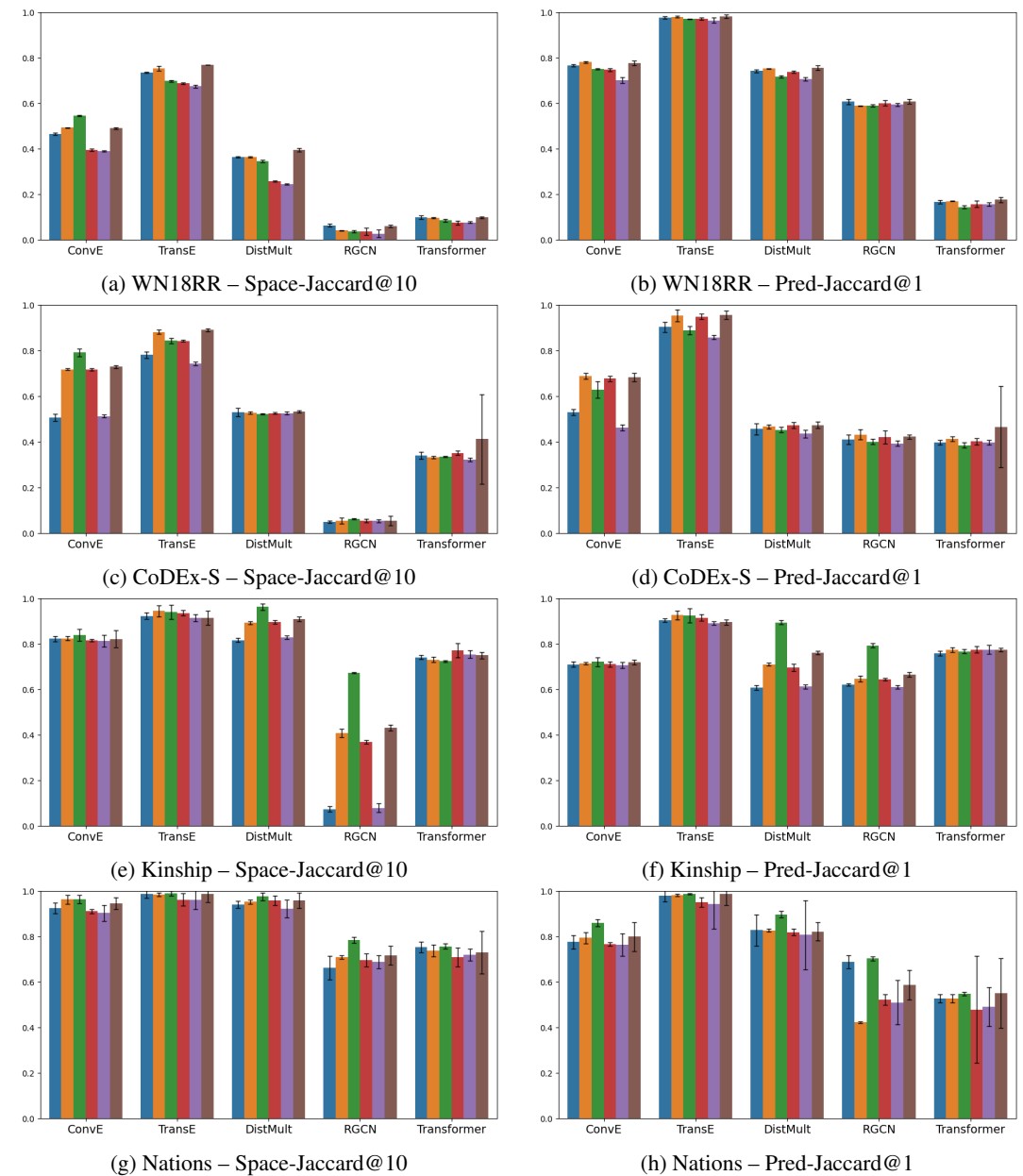

Figure 5: Impact of individual sources of randomness across datasets. Each source of randomness (including hardware) is varied independently while keeping all others fixed. Colors indicate the corresponding source of randomness: ■ seed_init, ■ seed_dropout, ■ seed_neg, ■ seed_order, ■ all, and ■ hardware. The category "all" includes all sources of randomness but hardware.

As for Jaccard, we define two stability measures based on RBO. At the prediction level, we average RBO scores over test queries:

$$\text{Pred-RBO@}K(I_1, I_2) = \frac{1}{|\mathcal{Q}_{\text{test}}|} \sum_{q \in \mathcal{Q}_{\text{test}}} \text{RBO}\big(\nu(I_1, q, K), \nu(I_2, q, K)\big). \quad (5)$$

At the embedding space level, we apply RBO to the neighborhoods of entities ranked by distance:

$$\text{Space-RBO@}K(E_1, E_2) = \frac{1}{|\mathcal{E}|} \sum_{e_i \in \mathcal{E}} \text{RBO}\big(\mathcal{N}_K^{E_1}(e_i), \mathcal{N}_K^{E_2}(e_i)\big). \quad (6)$$

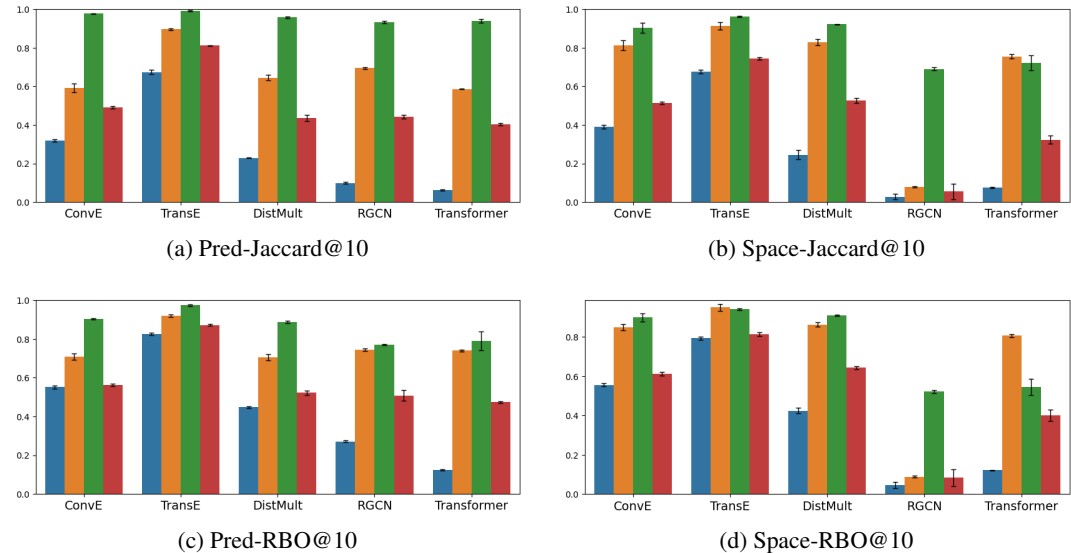

Figure 6: Comparison of prediction-level and space-level stability using Jaccard@10 and RBO@10 across models and datasets. The color codes represent the datasets: ■ WN18RR ■ Kinship ■ Nations ■ CoDEx-S

As shown in Figure 6, Jaccard and RBO yield very similar variation patterns across (model, dataset) pairs. On Nations, RBO slightly mitigates the score inflation caused by the small number of entities. The conclusions remain the same as with Jaccard: both prediction-level and space-level results reveal substantial instability.

## C.2 CENTERED KERNEL ALIGNMENT (CKA) (KORNBLITH ET AL., 2019)

We used Centered Kernel Alignment (CKA) as a complementary measure of space-level stability to validate our findings. Originally proposed to compare feature representations of examples across layers in neural networks, CKA relies on the Hilbert-Schmidt Independence Criterion to quantify the similarity between two representation spaces.

In our setting, we apply CKA directly to the entity embedding layers of different model instances, where entities take the role of examples. This allows us to assess whether different runs organize entities in a globally consistent manner.

Formally, given two embedding matrices $E^1, E^2 \in \mathbb{R}^{n \times d}$ for the same set of $n$ entities across two model instances, we compute their centered linear kernels $K_1^c$ and $K_2^c$, and define:

$$\text{CKA}(E^1, E^2) = \frac{\text{tr}(K_1^c K_2^c)}{\sqrt{\text{tr}(K_1^c K_1^c) \cdot \text{tr}(K_2^c K_2^c)}}. \tag{7}$$

A value close to 1 indicates that the two embedding spaces preserve highly similar global structures, whereas lower values reveal discrepancies in how entities are organized.

The CKA metric provides a measure of embedding similarity that does not rely on neighborhood structure (Figure 7c). When compared to Space-Jaccard@10 (Figure 7a), it avoids the score inflation observed on Nations due to its small number of entities. Aside from this difference, both metrics exhibit similar variation patterns across (model, dataset) pairs, supporting the view that they capture the same underlying notion of embedding space stability. Interestingly, CKA assigns consistently higher values (e.g. 0.8 to 1 for ConvE, TransE, and DistMult) which stands in slight tension with the lower stability suggested by Jaccard-based measures (Figures 7a and 8a). Still, RGCN and Transformer emerge as the least stable models, whereas TransE consistently appears as the most stable.

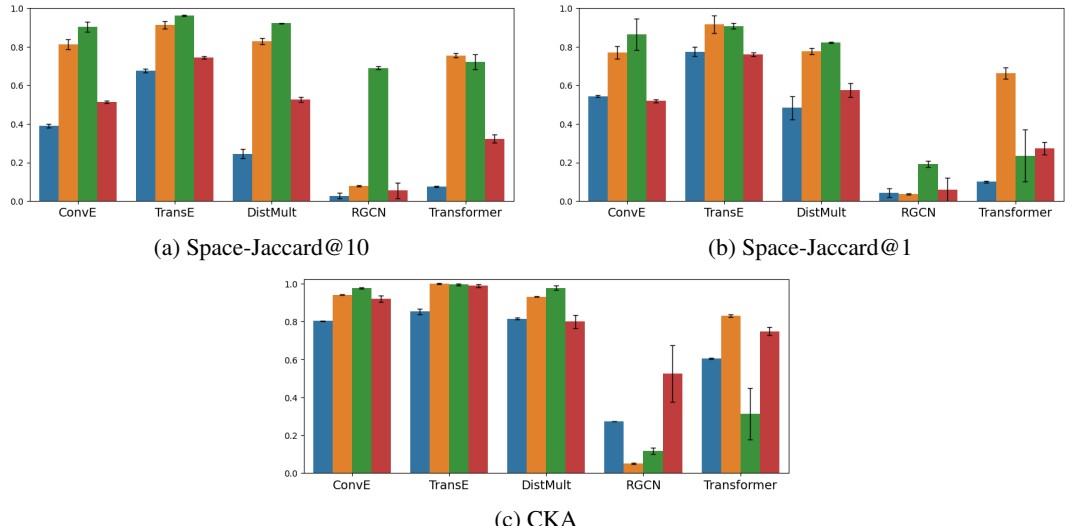

(a) Space-Jaccard@10

(b) Space-Jaccard@1

(c) CKA

Figure 7: Stability evaluation with complementary metrics across embedding space. The color codes represent the datasets: ■ WN18RR ■ Kinship ■ Nations ■ CoDEx-S

### C.3 LINK PREDICTION AMBIGUITY AND DISCREPANCY (ZHU ET AL., 2024)

We also used complementary prediction-stability notions that directly focus on the ground truth entity. Let $\Delta(I_1, I_2, q, K)$ be an indicator function that evaluates to 1 if both instances $I_1$ and $I_2$ either rank the correct entity for query $q \in \mathcal{Q}_{\text{test}}$ within the top-$K$ predictions or both rank it outside the top-$K$; otherwise, it returns 0.

We adapt the metrics of Ambiguity and Discrepancy introduced by Zhu et al. (2024). Their approach distinguishes a baseline model from other models that are $\epsilon$-close in terms of Hits@$K$. Instead, we compute these measures as pairwise aggregations over the entire group $G_{\mathcal{X}}^{\boldsymbol{C}}$, without referring to any specific baseline model. Furthermore, we do not enforce the models to be $\epsilon$-close in Hits@$K$.

Ambiguity measures the proportion of queries for which at least one pair of runs disagrees on the position of the ground truth entity:

$$\text{Amb@}K(G_{\mathcal{X}}^{\boldsymbol{C}}) = \frac{1}{|\mathcal{Q}_{\text{test}}|} \sum_{q \in \mathcal{Q}_{\text{test}}} \max_{I_1, I_2 \in G_{\mathcal{X}}^{\boldsymbol{C}}} \left(1 - \Delta(I_1, I_2, q, K)\right) \quad (8)$$

A high $\text{Amb@}K(G_{\mathcal{X}}^{\boldsymbol{C}})$ indicates that disagreements occur frequently across runs.

Discrepancy instead focuses on the worst-case perspective by measuring the maximum average disagreement between any two runs:

$$\text{Disc@}K(G_{\mathcal{X}}^{\boldsymbol{C}}) = \max_{I_1, I_2 \in G_{\mathcal{X}}^{\boldsymbol{C}}} \frac{1}{|\mathcal{Q}_{\text{test}}|} \sum_{q \in \mathcal{Q}_{\text{test}}} \left(1 - \Delta(I_1, I_2, q, K)\right) \quad (9)$$

A high $\text{Disc@}K(G_{\mathcal{X}}^{\boldsymbol{C}})$ means that there exists at least one pair of runs that highly disagrees on whether ground truth entities are ranked within the top-$K$ predictions.

Hence, Ambiguity and Discrepancy measure stability exclusively with respect to an agreement on the presence or absence of the ground truth in the top-$K$. This limitation can lead to issues when assessing the stability of models. To illustrate, consider the extreme case where two randomly initialized models are highly unlikely to rank the ground truth within the top-$K$. Here, the Ambiguity is reported as zero, although the models only agree on failing to predict the correct triple. To a lesser extent, this effect may also explain why Ambiguity@10 tends to be low across datasets (Figures 8d): if a triple is difficult to predict, most models will consistently exclude it from the top 10, which is then counted as agreement. Yet, this provides no insight into which triples populate the top 10 list, which are precisely the predictions of interest for KG completion. For this reason, we favor Jaccard similarity, as it directly evaluates the stability of the predicted top-$K$ candidates.

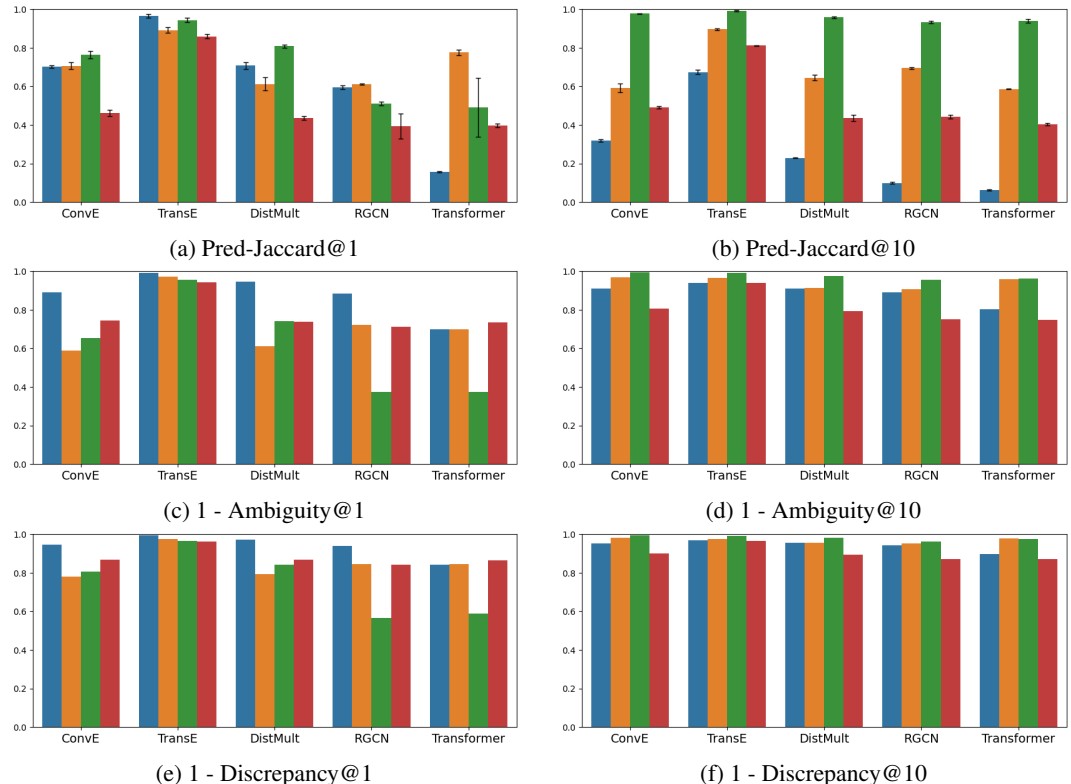

Figure 8: Stability metrics for predictions: Pred-Jaccard, $1 -$ Ambiguity, and $1 -$ Discrepancy. For all metrics, values closer to $1$ indicate stable models, while values closer to $0$ denote instability. The color codes represent the datasets: ■ WN18RR ■ Kinship ■ Nations ■ CoDEx-S

We argue that the discussed limitation of Ambiguity is reflected in our results. As shown in Figures 8c–8d, Ambiguity indicates higher stability on WN18RR than on other datasets, in contrast to Jaccard (Figures 8a–8b). This may be explained by the dominance of two relations: _hypernym (40.1%) and _derivationally_related_form (34.2%). According to Table 4 of Chen et al. (2021), _hypernym is difficult to rank correctly (MRR $= .144$), which may lead models to agree on not placing the gold entity in the top-$K$. Conversely, _derivationally_related_form is comparatively easy to rank (MRR $= .947$), which may result in consistent agreement in ranking it among the top-$K$.

Nevertheless, Ambiguity remains useful to attest instability in the predictions (Figures 8c–8d), and once again confirms the comparatively high stability of TransE. We can draw the same conclusions for Discrepancy (Figures 8e–8f), which can be seen as the maximal Ambiguity observed when comparing models pairwise.

