# OpenReview forum: "Link Prediction or Perdition: the Seeds of Instability in Knowledge Graph Embeddings"
_ICLR.cc/2026/Conference — ICLR 2026 Conference Withdrawn Submission_

### Official Review · Reviewer_vwJx · 2025-10-28

**Soundness:** 2
**Presentation:** 3
**Contribution:** 1
**Rating:** 4
**Confidence:** 4

**Summary:**

This paper investigates the stability of Knowledge Graph Embedding Models (KGEMs) under variations in random seeds and other stochastic factors during training. The authors propose using Jaccard similarity over top-K predictions (Pred-Jaccard@K) and embedding neighborhoods (Space-Jaccard@K) as complementary stability metrics. Their findings challenge the reliability of KG completion pipelines and call for more robust evaluation protocols that account for prediction-level consistency.

**Strengths:**

Given the increasing use of KGEMs in real-world applications, understanding seed-induced variability is essential for trustworthy deployment.

The authors isolate and evaluate five distinct sources of randomness, using multiple models, datasets, and stability metrics.

**Weaknesses:**

The proposed metrics Pred-Jaccard@K and Space-Jaccard@K do not consider the order of candidate entities, which means they still cannot address the potential issues in Figure 1.

While the paper includes representative models from major families, it omits some recent and widely used architectures, e.g., ComplEx and RotatE.

Some datasets used in the paper are also small and outdated (e.g., Nations: 14 entities; Kinship: 104 entities).

The paper is purely empirical. A discussion or even preliminary analysis of why certain models (e.g., TransE) are more stable than others would add significant value.

From my perspective, if we want, we can reproduce the results of most KGEMs precisely by replacing all the random modules with deterministic (pseudo-random) algorithms. In most cases, random doesn't mean unstable, just like LLMs. The authors may exaggerate the negative impacts of randomness in KGEMs.

**Questions:**

Please see Weaknesses.

---

### Official Review · Reviewer_ToXJ · 2025-10-31

**Soundness:** 3
**Presentation:** 3
**Contribution:** 1
**Rating:** 2
**Confidence:** 5

**Summary:**

Performs an experimental study to assess the stability of multiple common knowledge graph embeddings models (KGEM) w.r.t. to changing training seeds. This is done by fixuing seeds for various sources of randomness (initialization, batching, negative sampling, dropout), running the model, and assessing stability of the top-k predictions. My main concern with this paper is its very limited contribution (see W1).

**Strengths:**

S1. Provides additional experimental evidence for KGEM stability

**Weaknesses:**

W1. Limited contribution. Instability has been studied in prior work (as the paper clearly describes) and remedies such as ensembling have also been explored. The paper thus adds a data point to a well-studied problem, but no more.

W2. Metrics not fully convincing. The paper assesses stability by (i) similarity of top-k predictions and (ii) similarity of entity embeddings. I do not find (i) convincing since it includes "wrong" predictions and ultimately assesses whether the model is unstable in its wrong predictions. That's not a very interesting question, and prior work indeed focused on the correct predictions instead. To emphasize this point, suppose that a model ranks the correct answer at the first position, and all other answers in random order afterwards. I'd not consider such a model unstable. As for (ii), the similarity of entity embeddings is useful, but is likely dependent on which notion of neighborhood is being used (e.g., Euclidean? Cosine?) and how the KGEM actually "uses" these embeddings.

W3. The set of considered models is overly limited. Popular and well-performing KGEM such as ComplEx or RotatE or HAKE are missing.

**Questions:**

-

---

### Official Review · Reviewer_UujM · 2025-10-31

**Soundness:** 2
**Presentation:** 2
**Contribution:** 2
**Rating:** 2
**Confidence:** 5

**Summary:**

**Problem:** The paper investigates a methodological issue in the evaluation of Knowledge Graph Embedding Models (KGEMs), the standard method for link prediction (LP) in incomplete KGs. The core problem is a discrepancy between standard ML evaluation and common practice in the KGEM field. Standard procedure mandates reporting performance averaged over multiple runs with different random seeds, but KGEM literature often reports metrics (e.g., MRR, Hits@K) from a single "best" run. The paper's motivation is that these aggregate scores conceal "local" instabilities. It posits that two KGEM instances, trained with different seeds, could achieve nearly identical MRR scores while producing "different triple-level predictions" and learning "distinct embedding spaces." This issue raises concerns about the reliability and reproducibility of KG completion, as the specific facts inferred by a model could vary based purely on the random seed used during its training.

**Methodology:** A framework for quantifying instability.
- The work systematically investigates the impact of four primary sources of randomness: Parameter Initialization (I), Triple Ordering (O), Negative Sampling (N), and Dropout (D).
- To isolate sources of instability, a "comparison group" is defined as a set of model instances obtained by varying one of the four source of randomness---linked to different seeds, while keeping the others fixed.
- The core of the analysis is to measure the pairwise similarity between all runs within a group using two primary metrics:
   - Prediction Stability (Pred-Jaccard@K): Measures the Jaccard similarity between the sets of top-K predicted entities for the same query (e.g., (h, r, ?)), averaged over all test queries. A low score indicates the models are predicting different facts to complete the KG.
   - Embedding Space Stability (Space-Jaccard@K): Measures the Jaccard similarity of the K-Nearest-Neighbors for each entity's embedding, averaged over all entities. A low score indicates the models learned different geometric representations.
- The analysis is corroborated with additional similarity measures, i.e., Rank-Biased Overlap (RBO) and Centered Kernel Alignment (CKA).

**Experimental Setup:**
- 5 Models: TransE (geometric), DistMult (factorization), ConvE (convolutional), RGCN (GNN), and a Transformer-based model.
- 4 Datasets: WN18RR, CoDEx-S, Kinship, Nations.
- 4 GPUs: Nvidia 2080 Ti, 1080 Ti, T4, A40, A100.
- 5 seeds: 42, 283, 358, 698, 887.
- Hyperparameter search over embedding dimensions (128, 256, 512) and learning rates (10^{-6, -5, -4, -3, -2, -1}).

**Results:**
- *RQ1. Are KGEMs stable across different random seeds?*
   - No. The analysis reveals that while aggregate MRR scores are deceptively stable (exhibiting low standard deviation), the models show "pronounced variability" at the local level. Runs with similar MRR scores produce divergent top-K predictions (Pred-Jaccard@K) and learn inconsistent entity neighborhoods (Space-Jaccard@K).
- *RQ2. How do the different sources of randomness contribute to instability?*
   - All sources independently cause instability. The study shows that varying any single one of the four sources (I, O, N, or D) is sufficient to induce significant instability.
   - Running the exact same code and seeds on different GPU models produces variability of comparable magnitude to changing the software seeds.
- *RQ3. Is model stability correlated with link prediction performance?*
   - No. While the worst-performing models (lowest MRR) were, intuitively, highly unstable, the analysis found "no correlation between predictive performance and stability when comparing the best and median configurations." A model with a state-of-the-art MRR is not guaranteed to be more stable than a median-performing one, meaning performance and reliability are separate evaluation dimensions.

**Strengths:**

- Well-structured empirical analysis. The paper delivers a thorough, multi-dimensional analysis of stability in KGEMs, systematically isolating sources of randomness and including hardware effects.
- Practical implications. The observed disconnect between high MRR and local stability is a direct challenge to current benchmarking methodology.
- Clear experimental protocols; code, embeddings, and data availability is addressed, promoting transparent review.

**Weaknesses:**

- *Limited theoretical depth.* The discussion of why such instability arises is mostly speculative rather than formal. For instance, no mathematical analysis is given, even in outline, connecting model non-convexity, overparameterization, and instability. The lack of structural insight limits the generative potential for new stability-enforcing methods.
- *Scope of experimental coverage.* Only five KGEM models are analyzed, which—while representative—exclude many important recent advances, or those explicitly claiming robustness. The selected datasets (WN18RR, CoDEx-S, Kinship, Nations) are mostly small-sized and mid-sized benchmarks. As the authors themselves note, statistics—particularly on the smaller datasets—should not be broadly generalized to real-world settings. No results are shown for large-scale industry KGs or very large, noisy graphs, which are a key target application.
- *Inappropriate datasets for metrics.* The analysis relies on Jaccard@K for K=10. However, two of the four datasets, Nations and Kinship, are extremely small. Nations has only 14 entities in total. A Space-Jaccard@10 (K-Nearest-Neighbors at K=10) analysis on a graph of 14 nodes is statistically meaningless, as K represents ~70% of the entire dataset. The paper notes this in a figure caption ("scores... are inflated") but proceeds to include these invalid results in its main heatmaps (Fig 4), which average results and potentially skew the visual conclusions.
- *Limited bridging to solution space.* The paper identifies the problem with force but does not take the next step to propose, experiment with, or even outline concrete candidate mechanisms for stability.
- *Baseline settings.* Hyperparameter sweeps are relatively narrow. E.g., choices such as fixed dropout (except for seed variation) may confound variance attribution.
- *Uncontrolled implementation framework.* The study does not explicitly state that all 5 models (TransE, ConvE, RGCN, etc.) were implemented within a single, unified software library. The paper's appendix, in fact, references different sources (e.g., LibKGE, PyTorch Geometric) for different model configurations. This introduces a major confounding variable. The measured instability attributed to a model architecture may, in part, be an artifact of its specific library implementation.
- *Rank-agnostic prediction metric.* The primary metric for prediction, Pred-Jaccard@K, treats the top-K predictions as an unordered set. This is a poor fit for link prediction, where rank is a crucial factor. While the paper includes the rank-aware RBO metric in the appendix, its reliance on a rank-agnostic set metric for its main claims is questionable.
- *Contradiction in space metrics.* The paper's primary metric for embedding space, Space-Jaccard@K, shows very high instability (e.g., scores below 0.1 for RGCN). However, its secondary "validation" metric, CKA (Centered Kernel Alignment), shows very high stability (scores of 0.8-1.0) for the same models (e.g., TransE, ConvE). The paper dismisses this massive discrepancy by saying they show similar variation patterns, but the absolute scores tell different stories. More discussion is needed.

TYPOS AND SUGGESTIONS
- The figures appear to use default matplotlib or Draw.io color schemes. While the data is clear, the visual presentation does not match the quality of the research. Using more refined color palettes and ensuring high-resolution vector graphics would significantly improve the paper's presentation.
- There is a contradiction regarding the experimental hardware. Section 4.1 (L289-L290) states: "...the hardware H has been fixed to a GPU Nvidia GeForce RTX 2080 Ti (11 GiB)." Section 4.2.2 (L374-L377) states: "We run models on multiple GPUs: Nvidia GeForce RTX 2080 Ti... Nvidia GeForce GTX 1080 Ti... Nvidia Tesla T4... Nvidia A40... and Nvidia A100..." The authors should better clarify that the 2080 Ti was the default hardware for all experiments except for the specific hardware-variance analysis conducted for RQ2.
- Several acronyms are defined multiple times throughout the text, which disrupts the reading flow.
- The paper would benefit from a final proofread to catch minor errors, such as missing spaces after punctuation (e.g., noted by the reviewer on L301).

**Questions:**

The Reproducibility Statement (L492) promises to make code and embeddings publicly available. To adhere to best practices in open science and to provide clarity for future users, the authors should specify the open-source license (e.g., MIT, Apache 2.0) under which the materials will be released.

---

### Note · Authors · 2025-11-13

I have read and agree with the venue's withdrawal policy on behalf of myself and my co-authors.